# Irrelevance of linear controllability to nonlinear dynamical networks

Junjie Jiang[1] & Ying-Cheng Lai [1,2]

There has been tremendous development in linear controllability of complex networks. Real-world systems are fundamentally nonlinear. Is linear controllability relevant to nonlinear dynamical networks? We identify a common trait underlying both types of control: the nodal "importance". For nonlinear and linear control, the importance is determined, respectively, by physical/biological considerations and the probability for a node to be in the minimum driver set. We study empirical mutualistic networks and a gene regulatory network, for which the nonlinear nodal importance can be quantified by the ability of individual nodes to restore the system from the aftermath of a tipping-point transition. We find that the nodal importance ranking for nonlinear and linear control exhibits opposite trends: for the former large-degree nodes are more important but for the latter, the importance scale is tilted towards the small-degree nodes, suggesting strongly the irrelevance of linear controllability to these systems. The recent claim of successful application of linear controllability to *Caenorhabditis elegans* connectome is examined and discussed.

[1] School of Electrical, Computer and Energy Engineering, Arizona State University, Tempe, AZ 85287, USA. [2] Department of Physics, Arizona State University, Tempe, AZ 85287, USA. Correspondence and requests for materials should be addressed to Y.-C.L. (email: ying-cheng.lai@asu.edu)

In the development of a field that involves dynamical systems, when knowledge has accumulated to certain degree, the question of control would arise naturally. For example, in nonlinear dynamics, the principle of controlling chaos was articulated in 1990[1], after approximately a decade of intense research focusing on the fundamental understanding of chaotic dynamical systems. Likewise, in complex networks, the issue of control began to be addressed[2,3] also approximately after 10 years of tremendous growth of research triggered by the pioneering work on small world and scale-free networks. A key development is the systematic adoption of the linear structural controllability theory to complex networks with directed interactions[4]. Since then, there has been a great deal of effort in investigating the linear controllability of complex networks[5–21].

Control of linear dynamical systems is a traditional field in engineering[22,23]. Because of the simplicity in the possible dynamical behaviors that a linear dynamical system can generate (in contrast to nonlinear dynamical systems where the behaviors are extremely rich and diverse), the general objective is to design proper control signals to drive the system from an arbitrarily initial state to an arbitrarily final state in finite time. When applying the linear controllability theory to complex networks, a primary goal has been to determine the minimum number of controllers. This problem was addressed[4] for complex directed networks through the development of a minimum input theory based on the concept of maximum matching[24–26]. To generalize the linear controllability theory to networks of arbitrary structures (e.g., weighted or unweighted, directed or undirected), an exact controllability theory was developed[9] based on the Popov–Belevitch–Hautus (PBH) rank condition[27]. The exact controllability theory provides a computationally extremely efficient method to determine not only the minimum number of controllers but also the set of the nodes to which the control signals should be applied—the set of driver nodes, for complex networks of arbitrary topology and link structures[9].

The development of the linear controllability theories has played the role of stimulating research on controlling complex networks[28]. However, its limitations must not be forgotten. The fundamental assumption used in any linear controllability theory is that the nodal dynamics are described by a set of coupled linear, first-order differential equations. While such a setting may be relevant to engineering control systems, real-world systems are governed by nonlinear dynamics, such as biologically inspired networks[29]. In classical control engineering, it is well recognized that controllability for nonlinear systems requires a different set of tools to be developed compared to what is known for the controllability of linear systems[30]. A serious concern is the tendency to overstate the use or the predictive power of the linear controllability theories when they are applied to real-world physical or biological systems. For example, it was claimed recently[31] that linear network control principles can predict the neuron function in the *Caenorhabditis elegans* connectome, a highly nonlinear dynamical neuronal network. The goal of the present work is to legitimize this concern in a quantitative manner by presenting concrete and statistical evidence that linear network controllability may not be relevant to physically or biologically meaningful control of nonlinear networks.

The physical world is nonlinear. Network dynamics in biological or ecological systems are governed by nonlinear rules with no exceptions. Control of real world complex networks based on the rules of nonlinear dynamics has remained to be an extremely difficult problem. Existing strategies include local pinning[32–35], feedback vertex set control[36–38], controlled switch among coexisting attractors[39], or local control[21]. These methods belong to the category of open-loop control, i.e., one applies pre-defined control signals or parameter perturbations to a feedback vertex set

chosen according to some physical criteria. For certain nonlinear dynamical networks, especially those in ecology, closed-loop control can be articulated and has been demonstrated to be effective[40]. Recently, how to exploit biologically inspired agent-based control method to choose different alternative states in engineered multiagent network systems has been studied[41].

In order to answer the question "is linear controllability relevant to nonlinear dynamical networks?", two challenges must be met. Firstly, because of lack of general controllability framework for nonlinear networks, it is necessary to focus on *specific* contexts where nonlinear network control can be done in a physically or biologically meaningful way. We choose two such contexts: mutualistic networks in ecology[42–49] and a gene regulatory network from systems biology[50–52]. Secondly and more importantly, linear and nonlinear dynamical networks are fundamentally and characteristically different in many aspects, so are the respective control methods. How do we compare their control performances? (How can an apple be compared with a banana?) Our idea is that, even in the analog of apple–banana comparison, if one finds a common trait, e.g., the amount of sugar contained per gram of the substance, then a comparison between an apple and a banana in terms of the specific common trait is meaningful. We are thus led to seek a feature or a characteristic that is common in both nonlinear and linear network control. Specifically, we identify the statistical importance of individual nodes in control as such a common trait.

Our approach and main results can be described, as follows. Given a nonlinear dynamical network with its structure determined from empirical data, we focus on the concrete problem of harnessing a tipping point at which the system transitions from a normal state to a catastrophic state (e.g., massive extinction) or from a catastrophic state to a normal state abruptly as a system parameter changes through a critical point[45,49,53–59]. We exploit the ability of the individual nodes, via control, to make the system recover from the aftermath of a tipping point transition that puts the system in an extinction state. This enables a quantitative ranking of the importance of the individual nodes to be determined. The ranking is generally found to be linearly correlated with the nodal degree of the network, in agreement with intuition. The individual nodes, in terms of their ability to make the system recover, are drastically distinct. We then perform linear control on the same network by assuming artificial linear nodal dynamics. Using the exact controllability theory[9], we calculate the minimal control set. A key feature of linear network control, which was usually not emphasized in most existing literature on linear controllability[5–21] but was mentioned in a recent paper[60], is that the minimal control set of nodes is not unique. For a reasonably large network (e.g., of size of a few hundred), there can be vastly many such sets that are equivalent to each other in terms of control realization. Thus, in principle, there is a finite probability for a node in the network to be chosen as a control driver and the corresponding probability can be calculated from the ensemble of the minimal control sets. This probability can be defined as a kind of importance of the node in control relative to other nodes so that a nodal importance ranking can be determined. Because of the generality and universality of the linear control framework, the method to determine the nodal importance is applicable to any complex network. For a large number of real pollinator–plant mutualistic networks reconstructed from empirical data from different geographical regions of the world (Supplementary Table 2) and a representative gene regulatory network, we find that the linear importance ranking favors the small degree nodes, in stark contrast to the case of nonlinear control where large degree nodes are typically more valuable. The characteristic difference in the importance ranking of the nodes in terms of their role in control, linear or nonlinear, suggests that linear controllability may not be

relevant to physically or biologically justified nonlinear control for the mutualistic and gene regulatory networks.

## Results

**Irrelevance of linear controllability in a complex pollinator–plant mutualistic network**. The assumptions of this study are as follows. For linear dynamical networks, a general controllability framework exists, which can be used to determine the nodal importance ranking and is applicable to all networks. For nonlinear networks, because of the rich diversity in their dynamics, at the present a general control framework does not exist. The control strategy thus depends on the specific physical or biological context of the network.

To demonstrate the characteristic statistical difference between nonlinear and linear control, we take a representative pollinator–plant mutualistic network (network A), and calculate the node based, nonlinear and linear control importance according to Eqs. (1) and (7), respectively, as described in the "Methods" section. Figure 1 shows the 38 pollinator and plant species (Supplementary Table 1), together with the relative nonlinear and linear control importance as represented by the lengths of the green and blue bars beneath the images, respectively. There is a wide spread in the nonlinear control importance, but the linear control importance appears approximately uniform across the species. There are cases where a node is not important at all for nonlinear control (e.g., the first, fifth, and sixth species in the bottom row), but the node is important for linear control. The statistical characteristics of the nodal importance in nonlinear and linear control are thus drastically distinct. An examination of other empirical mutualistic systems reveals that, for some networks, the behaviors are similar to those in Fig. 1, while in others, the nodal importance shows opposite trends in nonlinear and linear control. For example, there are

cases where the nonlinear control importance tends to increase with the nodal degree, but the linear nodal importance shows the opposite trend. These results suggest that linear controllability may not be useful for controlling the actual nonlinear dynamical network.

**Nonlinear and linear control importance**. We present quantitative results of nonlinear and linear control importance for four empirical mutualistic networks described in the "Methods" section, as shown in Fig. 2. (Similar results from a large number of additional networks are presented in Supplementary Figs.) For a given empirical network, to calculate the nonlinear control importance based on definition (1), we begin from a zero value of the average mutualistic interaction strength $\gamma_0$, where the system is in an extinction state without control, apply the control by setting the abundance of a pollinator species at $A_S = 1.5$, and systematically increase the value of $\gamma_0$ towards a relatively large value (e.g., 3.0). During this process, the recovery point $\gamma_c^i$ can be obtained. When the values of the recovery point for all pollinator species have been calculated, Eq. (1) gives the control importance for each species, as shown in Fig. 2a–d for networks $A-D$, respectively, where the index of the pollinator species on the abscissa is arranged according to the nodal degree. Apart from statistical fluctuations, there is a high level of positive correlation between the nonlinear control importance and degree, i.e., larger degree nodes tend to be more important. In particular, managed control of larger degree nodes is more effective for species recovery. To obtain the linear control importance according to Eq. (7), we use 1000 random minimum controller sets as determined by the linear exact controllability to calculate the probability for each species to be chosen as a driver node. Note that, because of the artificial imposition of linear time-invariant dynamics on each node, there is a probability for any species to

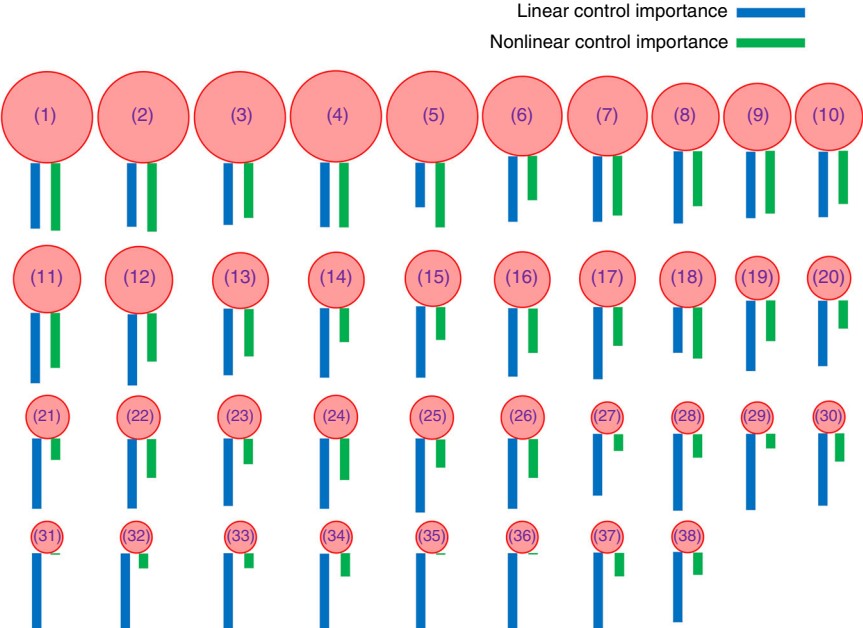

**Fig. 1** Distinct characteristics in nonlinear and linear control of a representative complex mutualistic network. The system is network A reconstructed based on empirical data from Tenerife, Canary Islands[67]. The numbers of pollinators, plants, and mutualistic links are $N_A = 38$, $N_P = 11$, and $L = 106$, respectively. For each node, the species name is given in Supplementary Table 1. The length of the green bar below each species is indicative of the relative importance of the node in tipping point control of the actual nonlinear dynamical network, which is calculated based on Eq. (1). The blue bars illustrate the relative importance of the nodes when the system is artificially treated as a linear, time-invariant network, which are calculated according to Eq. (7). There is great variation in the lengths of the green bars for different species, demonstrating a highly non-uniform nonlinear control importance ranking. In contrast, there is little variation in the length of the blue bars among the different species, indicating an approximately uniform linear control importance ranking. Linear controllability may thus not be useful for controlling the actual nonlinear dynamical network

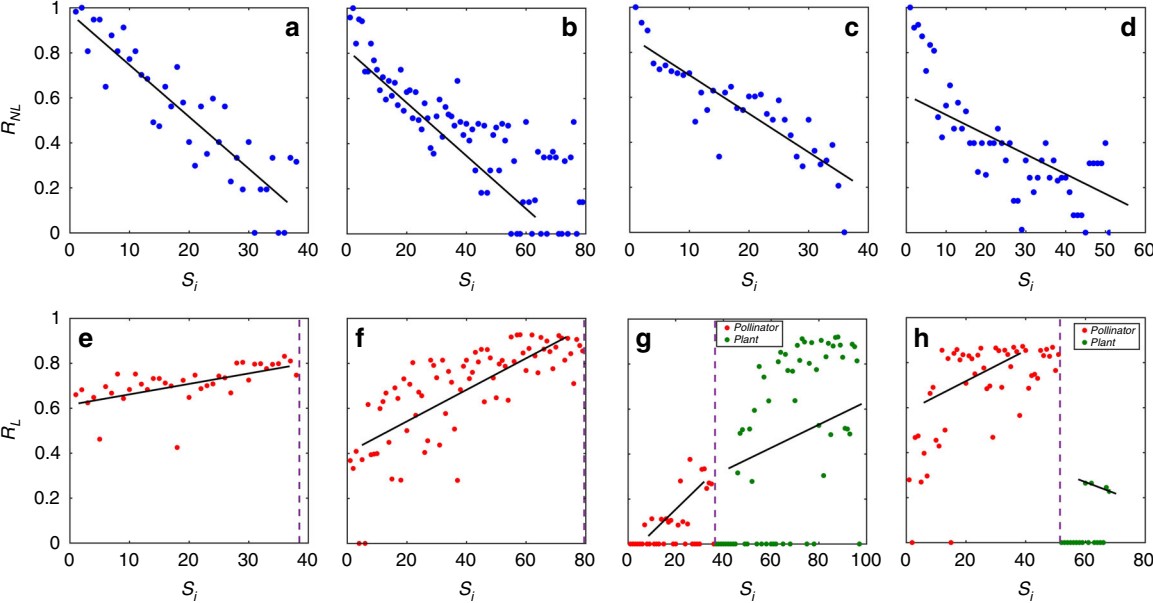

**Fig. 2** Contrasting behaviors of nodal importance ranking in nonlinear and linear control. The four empirical networks are labeled as *A*, *B*, *C*, and *D* with details given in the "Methods" section. **a–d** Nonlinear and **e–h** linear control importance ranking for networks *A–D*, respectively. For tipping point control of the nonlinear network in **a–d**, only the pollinator species are subject to external intervention through the managed maintenance of the abundance of a single species. The nodal index on the abscissa of each panel is arranged according to the degree ranking of the node: from high to low degree values (left to right). For the set of nodes with the same degree, their ranking is randomized. (The dependence of nonlinear and linear control importance on the actual degree value is presented in Supplementary Note 2.) The nonlinear control importance is calculated from Eq. (1) for the parameter setting $h = 0.2$, $t = 0.5$, $\beta_{ii}^{(A)} = \beta_{ii}^{(P)} = 1$, $\beta_{ij}^{(A)} = \beta_{ij}^{(P)} = 0$, $\alpha_i^{(A)} = \alpha_i^{(P)} = -0.3$, and $\mu_A = \mu_P = 0.0001$. The coupled nonlinear differential equations are solved using the standard Runge–Kutta method with the time step 0.01. The distinct feature associated with nonlinear control is that, in spite of the fluctuations, larger degree nodes tend to be more important (i.e., more effective in recovering the species abundances after a tipping point). The linear control importance ranking in **e–h** can be calculated for all species based on definition (7), because the corresponding artificial linear dynamical network does not distinguish between pollinator and plant species. In each panel, the pollinators (red dots) and plants (green dots) are placed on the left and right side, respectively, and are arranged in descending values of their degree, with a vertical dashed line separating the two types of species. The striking result is that, for the pollinators, their ranking of linear control importance exhibits a trend opposite to that of nonlinear control importance. A similar behavior occurs for ranking based on betweenness centrality and actual degrees (Supplementary Note 2 and Supplementary Figs. 3–5)

be a driver node, regardless of whether it is a pollinator or a plant species. The results are presented in Fig. 2e–h for networks *A−D*, respectively, where the linear control importance of the pollinators (red dots) and that of the plants (green dots)—separated by the vertical dashed line, are shown. The common feature among the four empirical networks is that the linear control importance ranking has an opposite trend to the nonlinear control importance ranking. That is, smaller degree nodes tend to be more important for linear control. The correlation between linear control importance and degree is thus negative, which is in stark contrast to the behavior of nonlinear control importance. Overall, Fig. 2a–h reveals that, for nonlinear control of tipping points, managing large degree nodes can be significantly more effective than harnessing small degree nodes, but for linear control of the same network, the large degree nodes play little role in control as they rarely appear in any minimum controller set.

The linear control importance measure, as defined in Eq. (7), is rooted in the fact that, in the linear controllability theory, typically there are many equivalent minimum controller sets[60]. It is useful to visualize such sets. Figure 3a exhibits a graphical representation of an empirical mutualistic network—network *E* described in the "Methods" section, where the pollinators (red dots) and plants (green dots) are arranged along a circle, and the size of a dot is proportional to the degree of the corresponding node. By definition, mutualistic interactions mean that there are no direct links between any pair of dots with the same color—any link in the network must be between a red and a green dot. For this network, there are altogether $\sim 10^{12}$ minimum controller sets

of exactly the same size—three examples are shown in Fig. 3b–d, respectively, where the driver nodes are represented by black dots. A feature is that the minimum controller sets tend to avoid nodes of very large degrees in the network, which is consistent with the results in Fig. 2. The corresponding linear and nonlinear control importance rankings are shown in Fig. 3e, f, respectively. A comparison of these results indicates that the ranking behaviors are characteristically distinct, suggesting the difference between linear controllability and nonlinear control—the same message conveyed by Fig. 2 (and many additional examples in Supplementary Figs).

**Gene regulatory networks.** The opposite behaviors in the nodal importance ranking for linear controllability and nonlinear control also arise in gene regulatory networks. For such networks, tipping point dynamics similar to those in mutualistic networks can occur when a biological parameter is reduced, rendering feasible a similar control strategy (see "Methods"). Figure 4 shows, for the network of *S. cerevisiae* described in "Methods", the nonlinear and linear control importance rankings for two subnetworks: the giant component (Fig. 4a, c) and the subnetwork of all nodes with input connections (Fig. 4b, d). Because of the dense connectivity in the giant component subnetwork, for linear control the size of the minimum controller set is $N_D = 4$ (Fig. 4c). For the subnetwork in (Fig. 4b, d), we have $N_D = 17$. Note that, for nonlinear control of the subnetwork (Fig. 4b), there are several genes that have zero nonlinear control

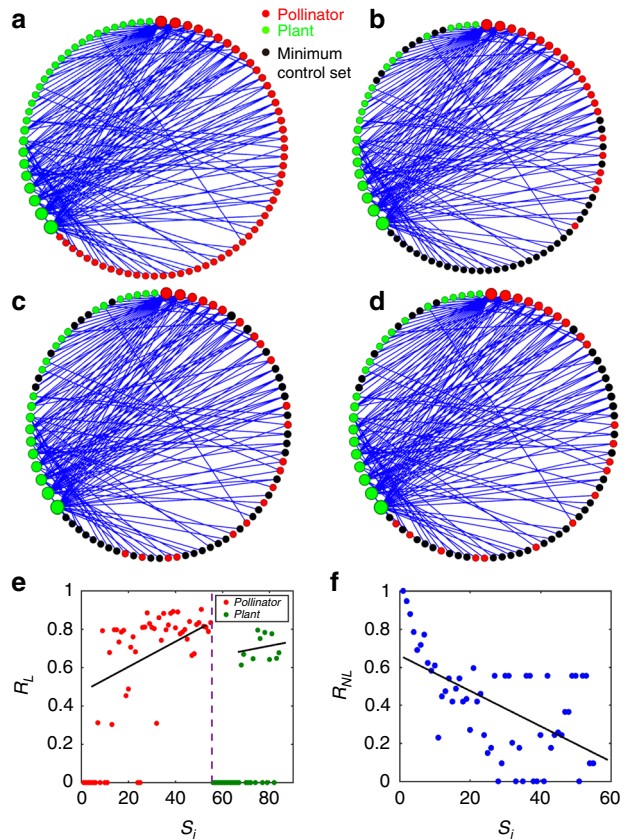

**Fig. 3** Examples of distinct minimum controller sets associated with linear control. For mutualistic network E as described in Methods, **a** network structure, where the size of a circle (red and green for a pollinator and a plant, respectively) is proportional to the degree of this node, **b**–**d** three examples of minimum controller sets (black dots), **e** linear control importance ranking, and **f** nonlinear control importance ranking. Other parameters are the same as those in Fig. 2

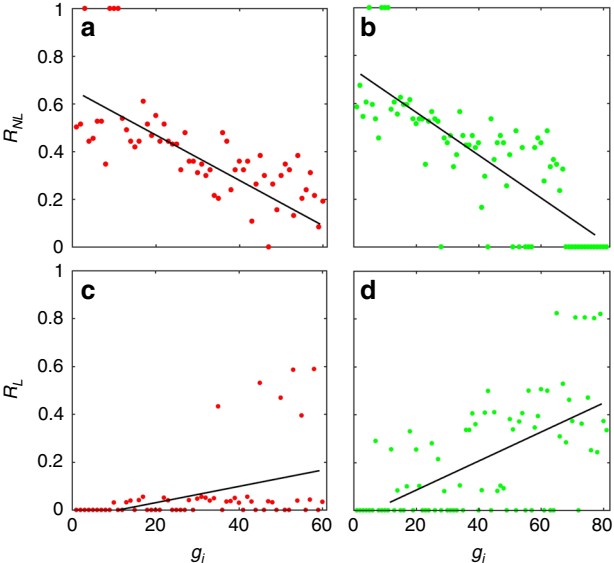

**Fig. 4** Nodal importance rankings associated with nonlinear and linear control of a gene regulatory network. **a**, **c** Nonlinear and linear control importance rankings for the subnetwork of the giant component of size 60, respectively. In **a**, there are four nodes with $R_{NL} = 1$, because controlling any of these genes will make the system recover immediately from the tipping point collapse when the direction of the change in the bifurcation parameter is reversed. For linear control in **c**, the size of any minimum controller set is $N_D = 4$. **b**, **d** Nonlinear and linear nodal importance rankings for the subnetwork of 81 nodes with input connection. In **b**, there are several genes with $R_{NL} = 0$, as each gene in this group lacks the ability to restore the entire system even when its activity level is maintained at a high level through external control. For linear control in **d**, any minimum controller set has $N_D = 17$ nodes. In all panels, the nodal index along the abscissa is arranged in the descending order of the outgoing degrees of the genes

importance, i.e., external management of the activation level of any of these genes is unable to restore the network function destroyed by a tipping point transition. The striking finding is that, for linear control, these genes are exceptionally important because the probability for any of these genes to belong to a minimum controller set is disproportionally high (e.g., >80%). If one follows the prediction of the linear controllability theory to identify those nodes as important and attempts to use them as the relevant nodes for actual control of the nonlinear network, one would be disappointed as harnessing any of these genes will have no effect on the tipping-point dynamics of the network. The occurrence of such genes with zero nonlinear control importance is the result of the interplay between the Holling-type of non-linear dynamics and the complex network structure.

**Pearson correlation and cosine distance**. For the five mutualistic networks ($A - E$) and two gene regulatory subnetworks tested so far, the correlation between nonlinear and linear control importance is negative, as shown in Figs. 2–4. To test if this holds for a broad range of empirical networks, we calculate the Pearson correlation and the cosine distance between linear and nonlinear control importance for a large number of real networks, as shown in Fig. 5. In most cases, the correlation is negative and the cosine distance is large. There are a few mutualistic networks with positive but small correlation. Out of the 43 mutualistic networks, only one has a large correlation value and a small cosine distance (one corresponding to the rightmost green circle). A peculiar

feature of this network is that it has only six pollinator species and any minimum controller set in linear control contains four such species, rendering atypical this case.

Our detailed comparison between the control importance ranking in a type of biologically meaningful nonlinear control and in linear control for a large number of real pollinator–plant mutualistic networks and a gene regulatory network provides evidence that linear controllability may generate results that are drastically inconsistent with nonlinear dynamical behaviors and control of the system. In no way should this be a surprise, as the assumption of linear, time-invariant dynamics cannot be expected to hold for nonlinear dynamical networks in the real world. However, there is a recent tendency to apply the linear controllability framework to real-world nonlinear systems, such as the *C. elegans* connectome[31] and brain networks[61–64]. Although the linear control framework may provide insights into nonlinear dynamical networks under some specific circum-stances, controlling highly nonlinear dynamical networks is still an open problem at the present. Nonetheless, a thorough analysis of the linear controllability would give clues to its inappropriate-ness and likely failure in real world systems (see Supplementary Note 1 and Supplementary Figs. 1 and 2).

## Discussion
It is apparent that the assumption of linear, time invariant nodal dynamics is not compatible with natural systems in the real world that are governed by nonlinear dynamical processes. Why then study the linear controllability of complex networks? There were two reasons for this. Firstly, when the development of the field of

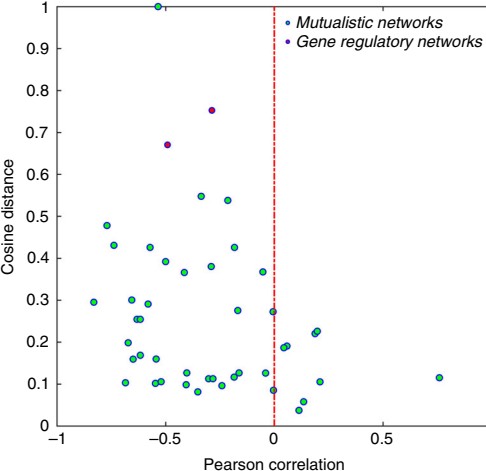

**Fig. 5** Pearson correlation and cosine distance between linear and nonlinear control importance. The abscissa and ordinate correspond to the values of Pearson correlation and cosine distance, respectively, between linear and nonlinear control importance. Each green circle corresponds to a real mutualistic network (there are 43 of them) and the two red dots are for the two gene regulatory subnetworks in Fig. 4. If there were a kind of relevance between nonlinear and linear control of the same network, the dots would concentrate in the lower right region of the plane with positive Pearson correlation and a small cosine distance. For most of the empirical networks tested, the dots are in the region of negative correlation with cosine distances below 0.4. For the two gene regulatory subnetworks, not only are the values of the Pearson correlation negative, the cosine distances are also large

complex network had reached the point at which the problem of control emerged as a forefront problem (around 2011), to adopt linear controllability, a well established framework in traditional control engineering, to complex networks seemed to be a natural starting point. The well-developed mathematical foundation of linear control made it possible to address the effect of complex network structure on the controllability in a rigorous manner[4,9], physical or biological irrelevance notwithstanding. Secondly, to study the linear controllability of complex networks is justified from the point of view of engineering, as linear dynamical systems are relevant to subfields in engineering, such as control and signal processing. That being said, the applicability of the linear controllability to real physical, chemical, and biological systems is fundamentally limited because of the ubiquity of nonlinear dynamics in natural systems—a well accepted fact, thanks to more than four decades of extensive and intensive study of nonlinear dynamics and chaos theory. It is imperative and a common sense understanding that the linear controllability of complex networks not be overemphasized and its importance and significance not be overstated.

Quite contrary to the common sense understanding, there are recent claims that linear network controllability is applicable to real biological systems[31,61–64] for gaining new understanding. Curiosity demands a thorough reexamination of these claims. More importantly, such claims, if they are indeed unjustified but remain uncorrected, can potentially generate undesirable and negative impacts on the further development of the field of complex network control. These considerations motivated our present work.

The main question we have set out to answer is whether linear controllability is actually relevant to controlling nonlinear dynamical networks. To be able to address this question, it is necessary to have nonlinear networked systems for which a certain type of physically or biologically meaningful control can be carried out.

We have identified two classes of such systems: complex pollinator–plant mutualistic networks in ecology and gene regulatory networks in systems biology. We focus on the physically significant issue of controlling tipping points, which enables the nodal importance in the control to be ranked. This is essentially a ranking associated with nonlinear control. Ignoring the nonlinear dynamics and simply using the network structure to treat it as a linear, time-invariant system enable us to calculate the minimum controller set in the linear controllability framework. Taking advantage of the exact controllability theory[9], we identify a large number of equivalent configurations of the minimum controller set and find that, typically, there is a probability for almost every node to be in such a set. This probability serves as the base for ranking the nodal importance in linear controllability. The two types of control importance rankings, one nonlinear and another linear, can then be meaningfully compared. The main finding of this paper is that the nonlinear and linear rankings are characteristically different for a large number of real world mutualistic networks and the gene regulatory network of *S. cerevisiae*. In particular, the nonlinear control importance ranking typically exhibits a behavior that in general favors high degree nodes. However, linear ranking typically exhibits the opposite trend that favors small degree nodes. These results are evidence that linear controllability theory generates information that is not useful for nonlinear control of tipping point dynamics in complex biological networks. A quite striking finding is that, for the gene regulatory network of *S. cerevisiae*, there are four genes with essentially zero nonlinear control importance in the sense that managed control of any of these genes is unable to recover the system from the aftermath of a tipping point transition. However, in linear control, these four genes are far more important than other nodes in the network. Thus, for the particular gene regulatory network studied here, linear controllability absolutely has nothing to do with the actual control of the nonlinear dynamical network.

In a recent work[31], it was claimed that linear structural controllability predicts neuron function in the *C. elegans* connectome. This real neuronal network has about 300 neurons, which contains four different types of neurons including the sensory neurons, inter-neurons, and motor neurons. A sensory neuron can generate an action potential propagating to other neurons, while an inter-neuron can receive action potentials from sensory neurons or other inter-neurons. The processes of generating and propagating action potentials are highly nonlinear. The claim of ref. [31] is thus questionable. We find that the *C. elegans* connectome, when artificially treated as a linear network, is uncontrollable if the control signals are to be applied to sensory neurons only. A calculation of the linear control importance reveals an approximately uniform ranking across all neurons. The surprising feature is that, on average, a muscle cell is almost twice as important as a motor neuron in terms of linear controllability, but biologically any control signal must flow from neurons to muscle cells, not in the opposite direction. Linear controllability thus yields a result that is apparently biologically meaningless. In fact, the ability to predict neuron function is based on signal propagation from some sensory to some motor neurons, which can be accomplished through random stimulation of some sensory neurons. Because of the existence of great many equivalent minimal control driver sets, which sensory neuron should be chosen to deliver a control signal is completely random. From the point of view of signal paths, there exist vastly large numbers of direct paths from the sensory to the motor neurons. Because of the approximately uniform ranking in nodal importance as a result of the existence of many equivalent minimum controller sets, linear controllability theory, when being used fairly in the sense of taking into considerations of the many controller set realizations, cannot possibly yield any path that is

more special than others to uncover hidden biological functions (see Supplementary Note 1). That is, it is not necessary to use linear controllability to predict any neuron function, contradicting the claim in ref. [31]. If control were to play a role in predicting some functions, it must be some kind of nonlinear control (which has not been achieved so far) due to the network dynamics' being fundamentally nonlinear.

Is it possible to use linear controllability as a kind of centrality measure for complex networks? The answer is "it depends." An essential requirement for such a measure is the ability to distinguish and rank the nodes in the network according to some criteria. Intuitively, one would hope that the nodes in the minimum controller set may be special and bear importance relative to other nodes. However, as demonstrated in our work, in a complex mutualistic network, the minimum controller set can be anything but unique. For a network of reasonable size, there is typically a vast number of equivalent configurations or realizations of the set, a fact that was seldom stated or studied in the existing literature of linear controllability of complex networks. We note that, besides the linear structural[4] and exact[9] controllability theories, there are alternative frameworks, such as the energy or linear Gramian-based controllability[61]. However, the Gramian matrix depends on the chosen minimum controller set and the control signal input matrix. Our finding that, for some networks, almost all nodes can be in some realizations of the minimum controller set with approximately equal probability makes it difficult to use or exploit linear controllability as a centrality measure for nodal ranking, such as network $A$ in Fig. 2e. However, for other networks, some nodes are always or never in a driver set, which give a distribution of nodes in the minimum controller set. The distribution with respect to the topology of the network may be informative and characteristic of some empirical contexts[11,60].

The type of nonlinear control exploited in this paper for comparison with linear controllability is controlled management of the aftermath of a tipping point transition to enable species recovery. While this is a special type of control, its merit is rooted in the feasibility to quantify and rank the ability of individual nodes to promote recovery of the nonlinear dynamical network, so that the node-based, nonlinear control importance can be meaningfully compared with the corresponding linear control importance. Is there a more general approach to nonlinear network control which can be used for comparison with linear network control? We do not have an answer at the present, as the collective behaviors of nonlinear dynamical networks are extremely diverse, so are the possible control strategies[21,32–39]. However, regardless of the type of nonlinear control, heterogeneity in the nodal importance ranking can be anticipated in general, due to the interplay between the nonlinear nodal dynamics and network structure. In contrast, as demonstrated in this paper, nodal importance ranking associated with linear controllability of complex networks exhibits a kind of heterogeneity opposite to that with nonlinear control, rendering linear controllability not useful for nonlinear dynamical networks in general.

## Methods

**General principle**. To obtain a statistical description of the roles played by the individual nodes and compare the nodal importance for nonlinear and linear control, we seek real world systems that meet the following two criteria: (a) the underlying dynamical network is fundamentally nonlinear, for which a detailed mathematical description of the model is available, and (b) there exists an issue of practical significance, with which nonlinear control is feasible. We find that mutualistic networks with a Holling type of dynamics[65,66] in ecology[42–49] and gene regulatory networks with Michaelis–Menten type of dynamics in systems biology[50–52] satisfy these two criteria, with respect to the significant and broadly interesting issue of controlling tipping points. The detailed models of the two types of networks are presented in Supplementary Note 3.

**Nonlinear dynamical networks**. We have performed calculations and analyses for a large number of real-world pollinator–plant mutualistic networks available from the Web of Life database (http://www.web-of-life.es), which were reconstructed from empirical data collected from different geographic regions across different continents and climatic zones. The results reported in the main text are from the following five representative mutualistic networks: (a) network $A$ ($N_A = 38$ and $N_P = 11$ with the number of mutualistic links $L = 106$) from empirical data from Tenerife, Canary Islands[67], (b) network $B$ ($N_A = 79$, $N_P = 25$, and $L = 299$) from Bristol, England[68], (c) network $C$ ($N_A = 36$, $N_P = 61$, and $L = 178$) from Morant Point, Jamaica[69], (d) network $D$ ($N_A = 51$, $N_P = 17$, and $L = 129$) from Tenerife, Canary Islands, and (e) network $E$ ($N_A = 55$, $N_P = 29$, and $L = 145$) from Garajonay, Gomera, Spain. Results from additional such networks are presented in Supplementary Figs. 6–13.

As a concrete example of gene regulatory networks, we study the transcription network of *S. cerevisiae* of 4441 nodes, for the representative parameter setting[51] $B = 1$, $f = 1$, and $h = 2$. In spite of the large number of genes involved in the network, the giant connected component in which each node can reach and is reachable from others along a directed path has 60 nodes only, and the size of the component in which each and every node has at least one incoming connection is 81.

**Nonlinear control importance ranking**. For convenience, here we use the term "nonlinear control importance" to mean the statistical characterization of the nodal importance when carrying out a physically meaningful type of control of the nonlinear dynamical network. Especially, we focus on controlling tipping points in complex pollinator–plant mutualistic networks and gene regulatory networks.

For the mutualistic networks, a typical scenario for a tipping point to occur is when the average mutualistic strength $\gamma_0$ is decreased towards zero. The tipping point occurs at a critical value $\gamma_0^c$, at which the abundances of all species decrease to near zero values. There is global extinction for $\gamma_0 \leq \gamma_0^c$. When $\gamma_0$ is increased from a value in the extinction region (e.g., in an attempt to restore the species abundances through improvement of the environment), recovery is not possible without control. A realistic control strategy was articulated, in which the abundance of a single pollinator species is maintained at a constant value, say $A_S$, through external means such as human management. We have observed numerically that, in the presence of control, a full recovery of all species abundances can be achieved—the phenomenon of "control enabled recovery." For the same value of the controlled species level $A_S$, the critical $\gamma_0$ value of the recovery point depends on the particular species (node) subject to control. A smaller recovery point in $\gamma_0$ thus indicates that the control is more effective, which is species dependent. The species, or nodes in the network, can then be ranked with respect to the control. This provides a way to define the nodal importance associated with control of the underlying nonlinear network. In particular, let $\gamma_c^i$ be the system recovery point when the $i$th pollinator is subject to control. Choosing each and every pollinator species in turn as the controlled species, we obtain a set of values of the recovery point: $\{\gamma_c^i\}_{i=1}^{N_A}$. Let $\gamma_c^{\max}$ and $\gamma_c^{\min}$ be the maximum and minimum values of the set. The importance of the pollinator species $i$ associated with control of the tipping point can then be defined as

$$R_{NL}^i = \frac{\gamma_c^{\max} - \gamma_c^i}{\gamma_c^{\max} - \gamma_c^{\min}}, \qquad (1)$$

where $0 \leq R_{NL}^i \leq 1$ and the control is more effective or, equivalently, the node subject to the control is more "important" if its corresponding value of $R_{NL}^i$ is larger.

For the gene regulatory network, decreasing the value of the bifurcation parameter $C$ from one will result in a tipping point at which the activities of all genes suddenly collapse to near zero values. The behavior of sudden extinction at the tipping point can be harnessed by maintaining the activity level of a single active gene, e.g., the most active gene. In particular, when such control is present, the genes "die" in a benign way in that the death occurs one after another as the value of $C$ approaches zero, effectively eliminating the tipping point. We also find that, without control, it is not possible to recover the gene activities by increasing the value of $C$, but a full recovery can be achieved with control. When a different gene is chosen as the controlled target, for the same level of maintained activity, the recovery point on the $C$-axis, denoted as $C_c$, is different, which provides the base to rank the "importance" of the genes with respect to control of the nonlinear network. A gene with a relatively smaller value of $C_c$ is more important, as control targeted at it is more effective to restore the gene activities in the network.

Similar to our approach to ranking the control importance for the pollinator–plant mutualistic networks, we define the following importance measure for gene $i$:

$$R_{NL}^i = \frac{C_c^{\max} - C_c^i}{C_c^{\max} - C_c^{\min}}, \qquad (2)$$

where $C_c^i$ is the critical expression level to recover the whole system when the gene is subject to control, $C_c^{\max}$ and $C_c^{\min}$ are the maximum and minimum values of the recovery point among all the genes in the network.

**Linear control importance ranking**. Here, the term "linear control importance ranking" is referred to as the statistical ranking of the nodes in terms of their roles in the control of the underlying linear dynamical network. This ranking can be determined by the exact linear controllability theory[9]. To do so, we follow the existing studies that advocate the use of linear controllability for real world networked systems, such as those in refs. [31,61–64]. That is, we completely ignore the fact that the mutualistic network system and the gene regulatory network are highly nonlinear dynamical systems and instead treat them fictitiously as linear dynamical networks. For a network of $N$ nodes whose connecting topology is characterized by the adjacent matrix $\mathcal{A}$, the linear control problem is formulated according to the following standard setting of canonically linear, time-invariant dynamical system:

$$\frac{d\mathbf{x}(t)}{dt} = \mathcal{A} \cdot \mathbf{x}(t) + \mathcal{B} \cdot \mathbf{u}(t), \qquad (3)$$

where $\mathbf{x}(t) \equiv (x_1(t), \ldots, x_N(t))^{\mathrm{T}}$ is the state vector of the system, $\mathcal{B}$ is the $N \times M$ input matrix ($M \le N$) that specifies the control configuration—the set of $M$ nodes (driver nodes) to which external control signals $\mathbf{u}(t) = (u_1(t), \ldots, u_M(t))^{\mathrm{T}}$ should be applied. In general, the linear networked system Eq. (3) can be controlled[70] for properly chosen control vector $\mathbf{u}$ and for $M \ge N_{\mathrm{D}}$, where $N_{\mathrm{D}}$ is the minimum number of external signals required to fully control the network. The classic Kalman controllability rank condition[22] states that, system Eq. (3) is controllable in the sense that it can be driven from any initial state to any desired final state in finite time if and only if the following $N \times NM$ controllability matrix

$$\mathcal{C} = (\mathcal{B}, \mathcal{A} \cdot \mathcal{B}, \mathcal{A}^2 \cdot \mathcal{B}, \ldots, \mathcal{A}^{N-1} \cdot \mathcal{B}),$$

has full rank:

$$\mathrm{rank}(C) = N.$$

For a complex directed network, the linear structural controllability theory[23] can be used to determine $N_{\mathrm{D}}$ through identification of maximum matching[4], the maximum set of links that do not share starting or ending nodes. A node is matched if there is a link in the maximum matching set points at it, and the directed network can be fully controlled if and only if there is a control signal on each unmatched node, so $N_{\mathrm{D}}$ is simply the number of unmatched nodes in the network.

An alternative linear controllability framework, which is applicable to complex networks of arbitrary topology (e.g., directed or undirected, weighted or unweighted), is the exact controllability theory[9] derived from the PBH rank condition[27]. In particular, the linear system Eq. (3) is fully controllable if and only

if the following PBH rank condition

$$\mathrm{rank}(c\mathcal{I}_N - \mathcal{A}, \mathcal{B}) = N, \qquad (4)$$

is met for any complex number $c$, where $\mathcal{I}_N$ is the $N \times N$ identity matrix. For any complex network defined by the general interaction matrix $\mathcal{A}$, it was proven[9] that the network is fully controllable if and only if each and every eigenvalue $\lambda$ of $\mathcal{A}$ satisfies Eq. (4). For a set of control input matrices $\mathcal{B}$, $N_{\mathrm{D}}$ can be determined as $N_{\mathrm{D}} = \min\{\mathrm{rank}(B)\}$. An equivalent but more practically useful criterion[9] is that, for a directed network, $N_{\mathrm{D}}$ is nothing but the maximum geometric multiplicity $\mu(\lambda_i)$ of the eigenvalue $\lambda_i$ of $A$:

$$N_{\mathrm{D}} = \max_i\{\mu(\lambda_i)\}, \qquad (5)$$

where $\lambda_i$ ($i = 1, \ldots, l \le N$) are the distinct eigenvalues of $\mathcal{A}$ and geometric multiplicity of $\lambda_i$ is given by

$$\mu(\lambda_i) = \dim V_{\lambda_i} = N - \mathrm{rank}(\lambda_i \mathcal{I}_N - \mathcal{A}).$$

For a directed network, the exact controllability theory gives the same value of $N_{\mathrm{D}}$ as determined by the structural controllability theory. For an undirected network with arbitrary link weights, $N_{\mathrm{D}}$ is determined by the maximum algebraic multiplicity (the eigenvalue degeneracy) $\delta(\lambda_i)$ of $\lambda_i$:

$$N_{\mathrm{D}} = \max_i\{\delta(\lambda_i)\}. \qquad (6)$$

An issue of critical importance to our work but which is often ignored in the existing literature on linear network controllability is the non-uniqueness of the set of the required driver nodes. In fact, for an arbitrary network with the value of $N_{\mathrm{D}}$ determined, there can be a large number of equivalent configurations of the driver node set. This can be seen from the matrix $c\mathcal{I}_N - \mathcal{A}$ that appears in the PBH rank condition Eq. (4). When $c$ is replaced by one of the eigenvalues of $\mathcal{A}$, say $\lambda_i$ (the one with the maximum algebraic multiplicity), the matrix $\lambda_i \mathcal{I}_N - \mathcal{A}$ contains at least one dependent row. The quantity $N_{\mathrm{D}}$ is nothing but the number of linearly dependent rows of $\lambda_i \mathcal{I}_N - \mathcal{A}$. The control signals should be applied to those nodes that correspond to the linearly dependent rows to make full rank the combined matrix $\lambda_i \mathcal{I}_N - \mathcal{A}, \mathcal{B}$ in Eq. (4), as illustrated in Fig. 6 for a small network of size $N = 10$. The key fact is that there can be multiple but equivalent choices of the linearly dependent rows of the matrix $\lambda_i \mathcal{I}_N - \mathcal{A}$. For the small $10 \times 10$ network in Fig. 6, there are 54 such choices. The $N_{\mathrm{D}} = 5$ driver nodes can then be chosen from the $N' = 9$ nodes as determined by the linearly dependent rows of $\lambda_i \mathcal{I}_N - \mathcal{A}$. When the network size $N$ is large, the $N_{\mathrm{D}} \ll N$ driver nodes can be chosen from $N' \lesssim N$ nodes. Since $N_{\mathrm{D}} \ll N'$, there can be great many distinct

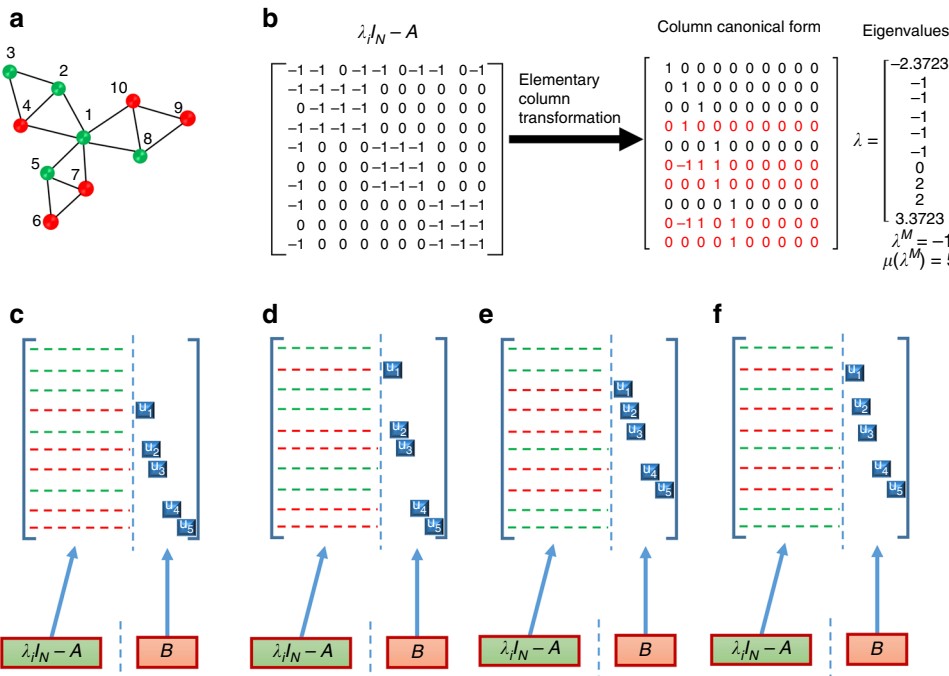

**Fig. 6** Illustration of non-uniqueness of driver node set in linear network control. **a** A 10-node undirected network. Five eigenvalues of the network connection matrix are identical: $\lambda = -1$, so its algebraic multiplicity is five. The geometric multiplicity of this eigenvalue is the number of linearly dependent rows in the matrix $\lambda_i \mathcal{I}_N - \mathcal{A}$, which can be determined through elementary column transforms. **b** The matrix $\lambda_i \mathcal{I}_N - \mathcal{A}$ and its representation after a series of elementary column transforms. The first, second, third, fifth, and eighth rows are distinct from all other rows, so they are linearly independent. The fourth, sixth, seventh, ninth, and tenth rows are linearly dependent rows. Because there are five linearly independent and five linearly dependent rows, the number of ways to choose the latter is 54. **c–f** Four distinct ways to choose the control input matrix to make the rank of the matrix $\lambda_i \mathcal{I}_N - \mathcal{A}, \mathcal{B}$ ten. For this small network of size ten, any one of the nine out of the the ten nodes can be chosen to be a driver node

possibilities for choosing the set of driver nodes (the number increases faster than exponential with the network size). It is thus justified to define the probability for a node to be chosen as one of the driver nodes, so that the importance of each individual node in linear control can be determined. Specifically, the linear control importance of node $i$ can be defined as

$$R_L^i = F_i/F, \qquad (7)$$

where $F$ is the total number of configurations of the minimum controller sets calculated and $F_i$ is the times that the $i$th node appears in these configurations. The probability $R_L^i$ thus gives the linear control importance ranking of the network, which can be meaningfully compared with the nonlinear control importance ranking.

**Reporting summary**. Further information on research design is available in the Nature Research Reporting Summary linked to this article.

## Data availability
All relevant data are available from the authors upon request.

## Code availability
All relevant computer codes are available from the authors upon request.

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

## Acknowledgements

We would like to acknowledge support from the Vannevar Bush Faculty Fellowship program sponsored by the Basic Research Office of the Assistant Secretary of Defense for Research and Engineering and funded by the Office of Naval Research through Grant no. N00014-16-1-2828.

## Author contributions

Y.-C.L. conceived the project. J.J. performed computations and analysis. Both analyzed data. Y.-C.L. wrote the paper with help from J.J.

## Additional information

**Competing interests:** The authors declare no competing interests.

