## [Peer Review File · Nature Communications]

Reviewers' comments:

Reviewer #1 (Remarks to the Author):

The authors present a very original study on the importance of different kinds of nodes for the controllability of complex networks. They distinguish linear vs. nonlinear control and uncover that nodes with low degree are most crucial for linear control in strong contrast to nonlinear control. They apply this theoretical finding for several real networks in ecology and genomics.

This is a very substantial study and it is very well presented. Therefore, I strongly recommend it for publication in Nat Comms.

However, I have some points which should be discussed in a minor revision:

- Here they refer to the degree of the nodes. It would be interesting to comment also another type of network characteristics, centrality measures, in particular betweenness from this perspective.
- Tipping points are here described as transition from a normal to a catastrophic state. This is not always the case (has not always to end in a catastrophic state).
- It would be important to describe the assumptions for this study a bit more explicitly or is the result valid for all networks?

Reviewer #2 (Remarks to the Author):

If the main message of the paper is just "linear controllability does not work for nonlinear systems", then such a conclusion is almost self-evident to control scientists and engineers. Classic control theory books, e.g. "Nonlinear dynamical control systems" by H. Nijmeijer and A.J. van der Schaft, Springer-Verlag 1990 and 2016, have extensive discussions on why controllability for nonlinear systems requires a different set of tools to be developed compared to what's known for controllability of linear systems. In fact, people have proposed concepts like accessibility using computational tools from differential geometry, to study nonlinear controllability. So I find this paper is rather just confirming what is well-known in the control field.

However, I do see that for physicists or some groups of researchers in the field of complex networks, nonlinear controllability may be an unfamiliar concept. People may have just used linear controllability to study nonlinear behaviors. Then the results in this paper, especially those two examples (one for pollinator-plant ritualistic networks and the other for gene regulatory networks) may of value.

So my overall conclusion is that if the targeted audience are people who are not familiar with nonlinear control theory, this paper has provided illustrative examples showing how the concept of controllability should be fundamentally different for linear and nonlinear systems. But for people who work in the field of nonlinear control, the results are not surprising and are in fact even obvious.

Reviewer #3 (Remarks to the Author):

Dear Editor,

In ms NCOMMS-19-13922, the authors consider the timely topic of controlling the dynamics of complex networks. More specifically, the manuscript presents a criticism of the use of the linear control framework in the context of a network whose dynamics are not inherently linear. The

authors' justification for this is based on an analysis of a node's importance in restoring network function after a "tipping point" that nominally leads to a mass extinction:

1. In a nonlinear framework, the nodes are ranked according to the "extinction threshold" below which the forced presence of the target species is insufficient to restore network function.
2. In a linear framework, the nodes are ranked according to the frequency with which they exist in the minimum sets of driver nodes.

The authors find that the nonlinear ranking is negatively correlated with node degree, whereas the linear ranking is positively correlated with degree. They therefore conclude that linear control is not an appropriate framework for considering nonlinear systems.

This strikes me as an interesting result, and I do not see any technical issues with the authors' analysis (though see comment 13, below). However, I have serious concerns about how the authors interpret their work. They cite several studies and write, "In no way should this be a surprise ... However, there is a recent tendency to overstate the applicability of the linear controllability framework to real world nonlinear systems ... Curiosity demands a thorough reexamination of these claims. More importantly, such claims, if found unjustified and uncorrected, can potentially generate undesirable and negative impacts on the further development of the field of complex network control."

I address this in more detail below (comment 11), but in short it seems to me that the authors are dismissing other studies as inherently flawed because of the findings in the present ms -- but without providing compelling and specific evidence that any of the conclusions reached in those papers are in fact flawed. In fact, it seems to me that the authors of those studies are aware of the advantages and disadvantages of the linear and nonlinear control framework, and that they carefully apply the linear framework in the context of their studies to reach their conclusions -- some of which are supported via experiment. Thus, it seems to me that the forceful criticism of the methodology that is offered in the present ms is weakly supported, despite the authors' forceful rhetoric. It seems to me that the authors need to support their criticisms by analyzing the specific methods and conclusions of those papers (which would certainly be a time-consuming task) to show how they are in fact flawed. Alternatively, the authors need to soften their criticism of those papers.

Elsewhere, I found some parts of the manuscript to be confusingly worded, and the coverage of the literature to be somewhat incomplete. I hope the authors find my detailed comments, below, to be useful.

Comment 1

In the abstract, the authors write, "In linear control, the importance can be characterized by the probability for a node to be in the minimum driver set." They make no similar statement explaining how they quantify node importance in the case of nonlinear networks. Thus, the following statement "We find that the rankings of nodal importance for nonlinear and linear control exhibit opposite trends..." is not as informative as it could be.

Comment 2

The third paragraph (The development of the linear controllability...) lays out the case that many dynamic systems are nonlinear and so assuming linear dynamics can be problematic. This is certainly a valid point, but it seems to me that the authors do a disservice to the broader literature that makes this argument; they only cite two papers (refs. 39-40). The following paragraph cites some additional papers in the context of describing existing methods for nonlinear control, but it seems to me that this is also somewhat cursory.

I encourage the authors to broaden the literature review here. They needn't feel compelled to cite any of the following papers in particular, but some that are relevant include (all DOIs):

1. 10.1109/TCNS.2018.2836303 (this is an introduction to a special journal issue on biological network control; papers in the issue may also be relevant)
2. 10.1103/PhysRevX.5.031036 (this paper leverages stochastic noise to exercise control in biomolecular networks)
3. 10.1063/1.4931570 (this review discusses nonlinearity in network control)
4. 10.1103/RevModPhys.88.035006 (a review of both linear and nonlinear control)

Furthermore, it is unclear to me what is meant by "simple kind of multistability" when the authors cite refs 45-49. Among the techniques discussed in these references is feedback vertex set control, which is in fact agnostic to many of the details of the network dynamics, linear or otherwise.

Comment 3

When the authors write, "In order to accomplish our goal..." on page 3, it is not immediately apparent from the preceding text what the goal is.

Comment 4

While I recognize the following text is from the introductory section, I found it to be confusingly vague:

"To develop a biologically viable control strategy to remove the tipping point so as to delay the occurrence of global extinction is of broad interest. We exploit controlling tipping points, which enables a quantitative ranking of the importance of the individual nodes in control to be determined. The ranking is generally found to be linearly correlated with that determined purely by the structure of the network, in agreement with intuition. The key issue is that the individual nodes, in terms of their ability to control the network tipping point, are drastically distinct. We then perform linear network control..."

Questions and comments on this text:

1. Sentence one says the goal is to "remove" the tipping point; sentence two refers to "controlling" them. Is the control objective their removal? This should be made explicit.
2. How is node importance determined from the structure of the network? There are many basic measures that are candidates, including degree and betweenness centrality, and many other more complicated measures that could be employed.
3. The nodes are distinct in what way? The last full sentence makes it unclear if they are distinct based on the first set of measures (control) or the second (network structure), or both.
4. If the authors "then" go on to linear network control, then the control mentioned initially is presumably nonlinear. What kind of nonlinear control was used? (This is important, cf Comment 1)

I recommend revising this text to address these questions.

Comment 5

The authors write, "A key feature of linear network control, which was usually not emphasized in the existing literature on linear controllability [7–28], is that the minimal control set of nodes is not unique." One exception is DOI 10.1038/srep46251, where the authors consider the relationship between the frequency of node presence in control sets and standard network measures. (See also the bottom of page 7, "...typically there are many equivalent minimum controller sets, which is seldom studied in the existing literature on linear network control.")

Comment 6

The species in Figure 1 should be identified by name (if not in the figure, then in the text, perhaps as SI).

Comment 7

If I follow the caption of Figures 2 and 4 correctly, the horizontal axis corresponds to the index of a species in a sorted list by node degree. This obfuscates the details of the actual degree distribution

and affects the nature of the fit. The authors should (1) make this explicit, (2) justify this choice, and (3) describe how their results would change if the fit was performed to the actual degree distribution.

Comment 8

On page 8, the authors write "Statistically, the ranking behaviors are characteristically distinct..." The meaning here is unclear. What statistical tests were done?

Comment 9

The authors frequently refer to the "irrelevance of linear controllability." I follow their argument but it seems to me this is not the best choice of terms. I would argue (to the benefit of the manuscript) that knowing precisely how the predictions of linear and nonlinear controllability differ can be informative in obtaining a holistic view of the network's behavior: they are only nonlinear to the extent that they differ from the expectations of a linear framework. So perhaps some rephrasing here would be appropriate.

Comment 10

On page 8 the authors write, "The occurrence of such genes with zero nonlinear control importance is the result of the interplay between the Holling-type of nonlinear dynamics and the complex network structure, to which the linear controllability theory is absolutely irrelevant." By the same logic in my previous comment, perhaps "...network structure, which does not influence the predictions of linear control."?

Comment 11

On page 9, the authors cite a few papers as over-relying on the linear framework: "However, there is a recent tendency to overstate the applicability of the linear controllability framework to real world nonlinear systems such as the *C. elegans* connectome [38] and brain networks [40, 72, 73]."

First, let me note that there is an apparent contradiction in regard to reference 40, because the authors cite the same paper earlier in the text: "If not careful, such claims could give the impression that linear network controllability theories are omnipotent [39, 40]..." My initial reading of this text was that the authors were pointing to references 39 and 40 as supporting the authors' point: it is best not to over-rely on linear controllability. However, on reading the text initially quoted above, it seems that perhaps they intend the opposite point -- that these papers are guilty of this appeal to omnipotence.

In any case, the text I initially quoted is unambiguous in characterizing the cited papers as over-relying on the linear framework. For this to be supported, it either needs to be self-evident from the papers that they use a linear framework in an inappropriate context, or the authors of the present ms must provide the supporting arguments themselves. On a quick review of each of these papers, it seems to me that the authors of the present ms are mischaracterizing each of them.

Indeed, each of the papers appears to (1) thoughtfully support the choice of employing a linear model in the context of the paper and/or (2) directly addresses the fact that the linear assumption is a simplification. For instance (I emphasize that these quotes are non-exhaustive):

From ref [38]:

"The nonlinearity of system (1) must be considered if we want to find out how to control the muscles. Here, however, we ask which neurons are necessary for control, which is defined by the controllability of the linearised system (2). Indeed, if (2) is locally controllable along a specific trajectory in state space, then the original nonlinear system (1) is also controllable along the same trajectory¹⁵. Furthermore, linear controllability predictions are consistent with simulations of neuronal networks with nonlinear dynamics^{16,17}."

From ref [39]:

"Although neural activity evolves through neural circuits as a collection of non-linear dynamic processes, these prior studies have demonstrated that a significant amount of variance in neural dynamics as measured by fMRI can be predicted from simplified linear models."

From the abstract of ref [40]:

"We conclude with a forward-looking discussion regarding how emerging results from network control -- especially approaches that deal with nonlinear dynamics or more realistic trajectories for control transitions -- could be used to directly address pressing questions in neuroscience."

From ref [72]:

"Decades of research demonstrate that neural dynamics are nonlinear. Yet, our approach is built on a linear model of these dynamics, and it is therefore imperative to delineate its strengths and weaknesses. First, we note that nonlinear behaviour may be accurately approximated by linear behaviour in certain scenarios..."

From ref [73]:

"We confirm the pragmatic utility of network control theory for nonlinear systems, extending previous work on linear approaches [30], and show..."

Now, the present manuscript certainly adds to the body of work characterizing the role of linear vs. nonlinear control, and it seems appropriate to me to present the results of the present ms as a cautionary tale for other researchers considering the use of linear controllability. But without additional support for the criticism of each of these papers, my reading of the current version of the manuscript is that it is excessively dismissive of the methods and findings of those papers, some of which support their predictions experimentally.

The sole possible exception is for ref [38], which the authors do investigate in their SI. Their main point is that linear controllability fails to differentiate between the node types (3 types of neurons and, separately, muscle cells). This is a valid criticism of the method -- it inherently considers only the wiring structure of the network -- and it falls to the researcher to identify biologically plausible control strategies. Indeed, the authors of [38] seem to my eye to take this approach. One can argue that a nonlinear approach is more realistic, and potentially more powerful, but in light of the experimentally supported findings reported in [38], it seems disingenuous to say (as the authors do in the SI) that the correlations they have found suggest the linear framework's "utter irrelevance to *C. elegans* connectome."

Comment 12

On page 12, the authors write "Our finding that almost all nodes can be in some realizations of the minimum controller set with approximately equal probability rules out any possibility of using or exploiting linear controllability as a centrality measure for the purpose of nodal ranking." While I agree with the broad point here -- that there are typically many minimal control sets and many nodes exist in many of them -- I think this is perhaps too strong. It is meaningful that some nodes are always or never in a driver set, and the distribution of these nodes in the topology of the network is informative and characteristic of empirical context (or the model used to build the network). See the authors' ref 15 and the above-mentioned DOI 10.1038/srep46251, written by some of the same authors.

Comment 13

The authors normalize their rankings according to the minimum and maximum thresholds observed for a given network. It seems to me that this ignores the range of values taken in a particular network: in one network the thresholds may be clustered near a low value, while in others they are clustered near a high value. It seems to me that something may be lost here, especially when comparing between the linear and nonlinear frameworks for a single network: a

broader range of values on an absolute scale makes the reported correlation more meaningful than if the range is very small.

While I do not expect this to dramatically impact the main results of the paper, it would be useful for the authors to provide some additional detail here.

Minor Comments

This is a quibble, but the sentence "Given a real world nonlinear network, we focus on the concrete problem of harnessing tipping point dynamics" strikes me as something of a contradiction in terms: a network is a model of a real-world system, with all of the trade-offs and design decisions that come along with the term "model." I suggest revision here.

The final sentence before the Results section is wordy and should be rewritten.

There is an unpaired left parentheses near the end of page 8: "(one corresponding..."

In the Methods, the statement "In the presence of control, a full recovery of all species abundances can be achieved (the phenomenon of "control enabled recovery")" should be supported with a citation.

Point-by-point response to reviewer comments and description of changes made

Reviewer #1

General Comment: *“The authors present a very original study on the importance of different kinds of nodes for the controllability of complex networks. They distinguish linear vs. nonlinear control and uncover that nodes with low degree are most crucial for linear control in strong contrast to nonlinear control. They apply this theoretical finding for several real networks in ecology and genomics.*

This is a very substantial study and it is very well presented. Therefore, I strongly recommend it for publication in Nat Comms.

However, I have some points which should be discussed in a minor revision:”

Response: We are grateful that the reviewer recommended our work for *Nature Communications*.

Minor Comment 1: *“- Here they refer to the degree of the nodes. It would be interesting to comment also another type of network characteristics, centrality measures, in particular betweenness from this perspective.”*

Response: We have followed reviewer’s suggestion to test betweenness centrality, a representative centrality measure, and obtained essentially the same results as for the degree. Especially, the betweenness centrality of a node is the number of shortest paths through this node. In a typical mutualistic network, the large degree nodes serve as the bridge between the pollinators and plants species. As a result, there is positive correlation between the degree centrality and the betweenness centrality. The new results have been included in Sec. 3 in Supporting Information.

Minor Comment 2: *“- Tipping points are here described as transition from a normal to a catastrophic state. This is not always the case (has not always to end in a catastrophic state).”*

Response: We agree with the reviewer that a tipping point does not always end in a catastrophic state. For example, depending on the direction of parameter change, it can also be interpreted as the transition from an extinction state to a normal state. We have modified the statement in Introduction (on page 3), which reads

- Our approach and main result can be described, as follows. Given a nonlinear dynamical network with its structure determined from empirical data, we focus on the concrete problem of harnessing a tipping point of the system. Generally, a tipping point is a point of no return at which the system transitions from a normal state to a catastrophic state (e.g., massive extinction) or from a catastrophic state to a normal state, in an abrupt manner, as a system parameter changes through a critical point [57, 61, 65-72]. In the former case where the tipping point

transition is undesired, to develop a biologically viable control strategy to remove the tipping point so as to delay the occurrence of global extinction is of broad interest. In the latter case where the transition can land the system in a normal state, to induce a tipping point to achieve global restoration is desired.

Minor Comment 3: “- *It would be important to describe the assumptions for this study a bit more explicitly or is the result valid for all networks?*”

Response: For linear dynamical networks, a general controllability framework exists and has been used in this study. Thus, the linear control methodology is universally applicable to all networks. For nonlinear networks, because of the rich diversity in their dynamics, at the present a general control framework does not exist. The control strategy thus depends on the specific physical or biological context of the network. For example, for complex mutualistic networks, we define the control importance measure in terms of the system’s ability to recover from the aftermath of a tipping point - control enabled species recovery. To address referee’s concern, we have added the following statements in Introduction (on page 3):

- The assumptions of this study are as follows. For linear dynamical networks, a general controllability framework exists, which can be used to determine the nodal importance ranking, which is applicable to all networks. For nonlinear networks, because of the rich diversity in their dynamics, at the present a general control framework does not exist. The control strategy thus depends on the specific physical or biological context of the network.

In addition, we have modified the pertinent statement in Introduction (lines 4-15 on page 4) to:

- For a reasonably large network (e.g., of size of a few hundred), there can be vastly many such sets that are equivalent to each other in terms of control realization. Thus, in principle, there is a finite probability for a node in the network to be chosen as a control driver and the corresponding probability can be calculated from the ensemble of the minimal control sets. This probability can be defined as a kind of importance of the node in control relative to other nodes so that a nodal importance ranking can be determined. Because of the generality and universality of the linear control framework, the method to determine the nodal importance is applicable to any complex network. For a large number of real pollinator-plant mutualistic networks reconstructed from empirical data from different geographical regions of the world and a representative gene regulatory network, we find that the linear importance ranking favors the small degree nodes, in stark contrast to the case of nonlinear control where large degree nodes are typically more valuable. The characteristic difference in the importance ranking of the nodes in terms of their role in control, linear or nonlinear, suggests that linear controllability may not be relevant to physically or biologically justified nonlinear control for the mutualistic and gene regulatory networks tested in this paper.

Referee #2

General Comment: *“If the main message of the paper is just ”linear controllability does not work for nonlinear systems”, then such a conclusion is almost self-evident to control scientists and engineers. Classic control theory books, e.g. ”Nonlinear dynamical control systems” by H. Nijmeijer and A.J. van der Schaft, Springer-Verlag 1990 and 2016, have extensive discussions on why controllability for nonlinear systems requires a different set of tools to be developed compared to what’s known for controllability of linear systems. In fact, people have proposed concepts like accessibility using computational tools from differential geometry, to study nonlinear controllability. So I find this paper is rather just confirming what is well-known in the control field.*

However, I do see that for physicists or some groups of researchers in the field of complex networks, nonlinear controllability may be an unfamiliar concept. People may have just used linear controllability to study nonlinear behaviors. Then the results in this paper, especially those two examples (one for pollinator-plant ritualistic networks and the other for gene regulatory networks) may of value.

So my overall conclusion is that if the targeted audience are people who are not familiar with nonlinear control theory, this paper has provided illustrative examples showing how the concept of controllability should be fundamentally different for linear and nonlinear systems. But for people who work in the field of nonlinear control, the results are not surprising and are in fact even obvious.”

Response: We appreciate that the reviewer shares with us the same concern that, in the field of complex networks, linear controllability theory may have been inappropriately used to draw conclusions from fundamentally nonlinear dynamical networks. We are glad that the reviewer considered our examples valuable. The book by Prof. Nijmeijer and Prof. van der Schaft has been cited in the revised paper (Ref. [40], in the third paragraph on page 2):

- While such a setting may be relevant to engineering control systems, real-world systems are governed by nonlinear dynamics such as biologically inspired networks [39]. In classical control engineering, it is well recognized that controllability for nonlinear systems requires a different set of tools to be developed compared to what is known for the controllability of linear systems [40].

At the present, there seems to be a trend of “blind” use of linear controllability theory in the field of complex networks by researchers not in engineering or mathematical control, who constitute the main audience of *Nature Communications*. The main purpose of our study to make them aware of the fundamental difference between linear and nonlinear control.

Referee #3

General Comment part 1: *“In ms NCOMMS-19-13922, the authors consider the timely topic of controlling the dynamics of complex networks. More specifically, the manuscript presents a criticism of the use of the linear control framework in the context of a network whose dynamics are not inherently linear. The authors’ justification for this is based on an analysis of a node’s importance in restoring network function after a ”tipping point” that nominally leads to a mass extinction:*

- 1. In a nonlinear framework, the nodes are ranked according to the ”extinction threshold” below which the forced presence of the target species is insufficient to restore network function.*
- 2. In a linear framework, the nodes are ranked according to the frequency with which they exist in the minimum sets of driver nodes.*

The authors find that the nonlinear ranking is negatively correlated with node degree, whereas the linear ranking is positively correlated with degree. They therefore conclude that linear control is not an appropriate framework for considering nonlinear systems.

This strikes me as an interesting result, and I do not see any technical issues with the authors’ analysis (though see comment 13, below). However, I have serious concerns about how the authors interpret their work. They cite several studies and write, ”In no way should this be a surprise ... However, there is a recent tendency to overstate the applicability of the linear controllability framework to real world nonlinear systems ... Curiosity demands a thorough reexamination of these claims. More importantly, such claims, if found unjustified and uncorrected, can potentially generate undesirable and negative impacts on the further development of the field of complex network control.”

I address this in more detail below (comment 11), but in short it seems to me that the authors are dismissing other studies as inherently flawed because of the findings in the present ms – but without providing compelling and specific evidence that any of the conclusions reached in those papers are in fact flawed. In fact, it seems to me that the authors of those studies are aware of the advantages and disadvantages of the linear and nonlinear control framework, and that they carefully apply the linear framework in the context of their studies to reach their conclusions – some of which are supported via experiment. Thus, it seems to me that the forceful criticism of the methodology that is offered in the present ms is weakly supported, despite the authors’ forceful rhetoric. It seems to me that the authors need to support their criticisms by analyzing the specific methods and conclusions of those papers (which would certainly be a time-consuming task) to show how they are in fact flawed. Alternatively, the authors need to soften their criticism of those papers.”

Response: We are grateful for reviewer’s generally positive evaluation of our work: “This strikes me as an interesting result, and I do not see any technical issues with the authors’ analysis”. We apologize for giving the impression that we are “forcefully” criticizing other researchers’ work in this area - that was certainly not our intention. As we stated in our response to Reviewer 2, we wish to make researchers who have been applying the linear controllability theory to complex networks to be aware

of the fundamental difference between linear and nonlinear control. We have followed the reviewer's kind suggestion to soften the pertinent statements throughout the paper.

General Comment part 2: *“Elsewhere, I found some parts of the manuscript to be confusingly worded, and the coverage of the literature to be somewhat incomplete. I hope the authors find my detailed comments, below, to be useful.”*

Response: We truly appreciate the reviewer's detailed comments, which have been fully implemented in the revised manuscript.

Comment 1: *“In the abstract, the authors write, “In linear control, the importance can be characterized by the probability for a node to be in the minimum driver set.” They make no similar statement explaining how they quantify node importance in the case of nonlinear networks. Thus, the following statement “We find that the rankings of nodal importance for nonlinear and linear control exhibit opposite trends...” is not as informative as it could be.”*

Response: We have improved the relevant statements in the Abstract:

- In linear control, the importance can be characterized by the probability for a node to be in a minimum driver set. We study two types of nonlinear networks: a large number of empirical pollinator-plant mutualistic networks and a gene regulatory network. For these nonlinear dynamical networks, the importance can be quantified by the ability of the individual nodes, with control, to restore the whole system from the aftermath of a tipping point transition. We find that the rankings of nodal importance for nonlinear and linear control exhibit opposite trends: for the former large degree nodes are more important but for the latter, the importance scale is tilted towards the small degree nodes. The implication is that linear controllability may not be useful for these systems.

Comment 2 part 1: *“The third paragraph (The development of the linear controllability...) lays out the case that many dynamic systems are nonlinear and so assuming linear dynamics can be problematic. This is certainly a valid point, but it seems to me that the authors do a disservice to the broader literature that makes this argument; they only cite two papers (refs. 39-40). The following paragraph cites some additional papers in the context of describing existing methods for nonlinear control, but it seems to me that this is also somewhat cursory.*

I encourage the authors to broaden the literature review here. They needn't feel compelled to cite any of the following papers in particular, but some that are relevant include (all DOIs):

- 1. 10.1109/TCNS.2018.2836303 (this is an introduction to a special journal issue on biological network control; papers in the issue may also be relevant)*
- 2. 10.1103/PhysRevX.5.031036 (this paper leverages stochastic noise to exercise control in biomolecular networks)*

3. [10.1063/1.4931570](https://doi.org/10.1063/1.4931570) (this review discusses nonlinearity in network control)

4. [10.1103/RevModPhys.88.035006](https://doi.org/10.1103/RevModPhys.88.035006) (a review of both linear and nonlinear control)”

Response: We appreciate the reviewer’s kind suggestion and have cited the papers in the revised manuscript ([39], [49], [79]).

Comment 2 part 2: “*Furthermore, it is unclear to me what is meant by ”simple kind of multistability” when the authors cite refs 45-49. Among the techniques discussed in these references is feedback vertex set control, which is in fact agnostic to many of the details of the network dynamics, linear or otherwise.*”

Response: We have removed the vague statement about multistability and have changed the part in Introduction (top of page 3) to

- So far, controlling complex networks with nonlinear dynamics has been limited to strategies such as local pinning [42-45], feedback vertex set control [46-48], controlled switch among coexisting attractors [49,50] where the goal is to drive the system from an undesired to a desired attractor, or local control [51]. These methods belong to the category of open-loop control, i.e., one applies pre-defined control signals or parameter perturbations to a feedback vertex set chosen according to some physical criteria.

Comment 3: “*When the authors write, ”In order to accomplish our goal...” on page 3, it is not immediately apparent from the preceding text what the goal is.*”

Response: We have changed the statement in Introduction (at the beginning of the second paragraph on page 3) to

- In order to answer the question “is linear controllability relevant to nonlinear dynamical networks?”, two challenges must be met. Firstly, while linear dynamical networks afford a general controllability framework, for nonlinear networks no general control framework has been available, yet. Our resolution is to focus on *specific* contexts where nonlinear network control can be done in a physically or biologically meaningful way.

Comment 4 part 1: “*While I recognize the following text is from the introductory section, I found it to be confusingly vague: ”To develop a biologically viable control strategy to remove the tipping point so as to delay the occurrence of global extinction is of broad interest. We exploit controlling tipping points, which enables a quantitative ranking of the importance of the individual nodes in control to be determined. The ranking is generally found to be linearly correlated with that determined purely by the structure of the network, in agreement with intuition. The key issue is that the individual nodes, in terms of their ability to control the network tipping point, are drastically distinct. We then perform linear network control...”*”

Questions and comments on this text:

1. Sentence one says the goal is to "remove" the tipping point; sentence two refers to "controlling" them. Is the control objective their removal? This should be made explicit."

Response: There are two kinds of transitions for complex mutualistic networks and the gene regulatory network studied: the collapse tipping point at which the system transitions from a normal state to an extinction state and the control enabled restoration point at which the system switches from an extinction to a normal state. Our goal is to assess individual nodes' ability to remove the former and induce the latter through control that maintains the abundance of a target node (species). We have made this explicit in the revised manuscript (last paragraph on page 3).

Comment 4 part 2: *"2. How is node importance determined from the structure of the network? There are many basic measures that are candidates, including degree and betweenness centrality, and many other more complicated measures that could be employed."*

Response: To see how nodal importance depends on the network structure, we have now calculated the importance with respect to three measures: degree ranking, the actual degree, and the betweenness centrality. The new results are presented in Sec. 3 in Supporting Information.

Comment 4 part 3: *"3. The nodes are distinct in what way? The last full sentence makes it unclear if they are distinct based on the first set of measures (control) or the second (network structure), or both."*

Response: The nodes are distinct in terms of their ability to make the system to recover from the aftermath of a tipping point transition that puts the system in an extinction state. We have modified the original statement to (toward the end of page 3)

- We exploit the ability of the individual nodes, via control, to make the system recover from the aftermath of a tipping point transition that puts the system in an extinction state. This enables a quantitative ranking of the importance of the individual nodes to be determined. The ranking is generally found to be linearly correlated with the node degree of the network, in agreement with intuition. The key issue is that the individual nodes, in terms of their ability to make the system recover, are drastically *distinct*.

Comment 4 part 4: *"4. If the authors "then" go on to linear network control, then the control mentioned initially is presumably nonlinear. What kind of nonlinear control was used? (This is important, cf Comment 1)*

I recommend revising this text to address these questions."

Response: We have used a pinning type of open-loop control to drive the system from an extinction state to a normal state through maintaining the abundance of a single pollinator species at an approximately constant level - see Methods. The relevant statements in Introduction (on page three) have been changed to

- In the former case where the tipping point transition is undesired, to develop a biologically viable control strategy to remove the tipping point so as to delay the occurrence of global extinction is of broad interest. In the latter case where the transition can land the system in a normal state, to induce a tipping point to achieve global restoration is desired. We exploit the ability of the individual nodes, via control, to make the system recover from the aftermath of a tipping point transition that puts the system in an extinction state. This enables a quantitative ranking of the importance of the individual nodes to be determined. The ranking is generally found to be linearly correlated with the node degree of the network, in agreement with intuition. The key issue is that the individual nodes, in terms of their ability to make the system recover, are drastically *distinct*.

Comment 5: *“The authors write, ”A key feature of linear network control, which was usually not emphasized in the existing literature on linear controllability [728], is that the minimal control set of nodes is not unique.” One exception is DOI 10.1038/srep46251, where the authors consider the relationship between the frequency of node presence in control sets and standard network measures. (See also the bottom of page 7, ”...typically there are many equivalent minimum controller sets, which is seldom studied in the existing literature on linear network control.”)”*

Response: We apologize for having missed this importance reference, which has been cited in the revised paper with the following modified statement:

- A key feature of linear network control, which was usually not emphasized in most existing literature on linear controllability [7-28] but was mentioned in a recent paper [73], is that the minimal control set of nodes is not unique.

On page 8, we have written

- The linear control importance measure, as defined in Eq. (11), is rooted in the fact that, in the linear controllability theory, typically there are many equivalent minimum controller sets [73].

Comment 6: *“The species in Figure 1 should be identified by name (if not in the figure, then in the text, perhaps as SI).”*

Response: The names of all species have now been listed in Table S1 of Supporting Information.

Comment 7: *“If I follow the caption of Figures 2 and 4 correctly, the horizontal axis corresponds to the index of a species in a sorted list by node degree. This obfuscates the details of the actual degree distribution and affects the nature of the fit. The authors should (1) make this explicit, (2) justify this*

choice, and (3) describe how their results would change if the fit was performed to the actual degree distribution.”

Response: We choose to use the indices of the species instead of the degree for the reason that the degrees of many species in a mutualistic network are exactly the same. For the species, nonlinear and linear control importance can be different. If degree is plotted on the horizontal axis, there are only a small number of distinct values. However, we find that using the actual degree values does not change the results. For example, for the pollinators, their ranking of linear control importance still exhibits a trend opposite to that of nonlinear control importance. Following the reviewer’s suggestion, we have provided new results of the importance ranking in terms of the degree value in Sec. 3 of Supporting Information. In the caption of Fig. 2 in the main text, the following statement has been added:

- The nodal index on the abscissa of each panel is arranged according to the degree ranking of the node: from high to low degree values (left to right). For the set of nodes with the same degree, their ranking is randomized. (The dependence of nonlinear and linear control importance on the actual degree value is presented in Sec. 3 of Supporting Information.)

Comment 8: *“On page 8, the authors write ”Statistically, the ranking behaviors are characteristically distinct...” The meaning here is unclear. What statistical tests were done?”*

Response: The statement has been modified to

- The corresponding linear and nonlinear control importance rankings are shown in Figs. 3(e) and 3(f), respectively. A comparison of the results indicates that the ranking behaviors are characteristically distinct, suggesting the difference between linear controllability theory and nonlinear control - the same message conveyed by Fig. 2 (and many additional examples in SI).

Comment 9: *“The authors frequently refer to the ”irrelevance of linear controllability.” I follow their argument but it seems to me this is not the best choice of terms. I would argue (to the benefit of the manuscript) that knowing precisely how the predictions of linear and nonlinear controllability differ can be informative in obtaining a holistic view of the network’s behavior: they are only nonlinear to the extent that they differ from the expectations of a linear framework. So perhaps some rephrasing here would be appropriate.”*

Response: We have removed the phrase “irrelevance” throughout the text and used terms such as “characteristic difference between linear and nonlinear control” and “linear controllability may not be useful for controlling the actual nonlinear dynamical network”, etc.

Comment 10: *“On page 8 the authors write, ”The occurrence of such genes with zero nonlinear control importance is the result of the interplay between the Holling-type of nonlinear dynamics and the complex network structure, to which the linear controllability theory is absolutely irrelevant.” By the same logic in my previous comment, perhaps ”...network structure, which does not influence the predictions of linear control.”?”*

Response: We have modified the statement to

- If one follows the prediction of the linear controllability theory to identify those nodes as important and attempts to use them as the relevant nodes for actual control of the nonlinear network, one would be disappointed as harnessing any of these genes will have no effect on the nonlinear dynamics of the network. The occurrence of such genes with zero nonlinear control importance is the result of the interplay between the Holling-type of nonlinear dynamics and the complex network structure.

Comment 11 part 1: *“On page 9, the authors cite a few papers as over-relying on the linear framework: ”However, there is a recent tendency to overstate the applicability of the linear controllability framework to real world nonlinear systems such as the C. elegans connectome [38] and brain networks [40, 72, 73].”*

First, let me note that there is an apparent contradiction in regard to reference 40, because the authors cite the same paper earlier in the text: ”If not careful, such claims could give the impression that linear network controllability theories are omnipotent [39, 40]...” My initial reading of this text was that the authors were pointing to references 39 and 40 as supporting the authors’ point: it is best not to over-rely on linear controllability. However, on reading the text initially quoted above, it seems that perhaps they intend the opposite point – that these papers are guilty of this appeal to omnipotence.

In any case, the text I initially quoted is unambiguous in characterizing the cited papers as over-relying on the linear framework. For this to be supported, it either needs to be self-evident from the papers that they use a linear framework in an inappropriate context, or the authors of the present ms must provide the supporting arguments themselves. On a quick review of each of these papers, it seems to me that the authors of the present ms are mischaracterizing each of them.

Indeed, each of the papers appears to (1) thoughtfully support the choice of employing a linear model in the context of the paper and/or (2) directly addresses the fact that the linear assumption is a simplification. For instance (I emphasize that these quotes are non-exhaustive):

From ref [38]:

”The nonlinearity of system (1) must be considered if we want to find out how to control the muscles. Here, however, we ask which neurons are necessary for control, which is defined by the controllability of the linearised system (2). Indeed, if (2) is locally controllable along a specific trajectory in state space, then the original nonlinear system (1) is also controllable along the same trajectory¹⁵. Furthermore, linear controllability predictions are consistent with simulations of neuronal networks with nonlinear dynamics^{16,17}.”

From ref [39]:

”Although neural activity evolves through neural circuits as a collection of non-linear dynamic pro-

cesses, these prior studies have demonstrated that a significant amount of variance in neural dynamics as measured by fMRI can be predicted from simplified linear models.”

From the abstract of ref [40]:

”We conclude with a forward-looking discussion regarding how emerging results from network control – especially approaches that deal with nonlinear dynamics or more realistic trajectories for control transitions – could be used to directly address pressing questions in neuroscience.”

From ref [72]:

”Decades of research demonstrate that neural dynamics are nonlinear. Yet, our approach is built on a linear model of these dynamics, and it is therefore imperative to delineate its strengths and weaknesses. First, we note that nonlinear behaviour may be accurately approximated by linear behaviour in certain scenarios...”

From ref [73]:

”We confirm the pragmatic utility of network control theory for nonlinear systems, extending previous work on linear approaches [30], and show...”

Now, the present manuscript certainly adds to the body of work characterizing the role of linear vs. nonlinear control, and it seems appropriate to me to present the results of the present ms as a cautionary tale for other researchers considering the use of linear controllability. But without additional support for the criticism of each of these papers, my reading of the current version of the manuscript is that it is excessively dismissive of the methods and findings of those papers, some of which support their predictions experimentally.”

Response: We appreciate the reviewer’s informative comment. The Review article on controlling dynamics in brain networks [1] (Ref. [79] in the original manuscript) gives a comprehensive review of the current research progress in controlling the brain network, which includes methods that treat the brain network as a linear dynamical network and nonlinear control methods as well. A message of the paper is that linear controllability is not omnipotent.

We note that, for Ref. [39] in the original manuscript [2], the reason to use linear controllability lies in “prior studies demonstrated that a significant amount of variance in neural dynamics as measured by fMRI can be predicted from the simplified linear model.” Reference 72 [3] and 73 [4] in our original manuscript state that “. . . First, we note that nonlinear behaviour may be accurately approximated by linear behavior . . .” and “. . . We confirm the pragmatic utility of network control theory for nonlinear systems, extending previous work on linear approaches, and show . . .”, respectively. It is difficult to assess the exact meaning of these statements under the circumstances of study. Thus our statements in the original manuscript may be too strong. Nonetheless, we feel that treating a nonlinear dynamical network as a linear network would be problematic in some (if not all) cases. It is the purpose of our study to warn researchers that using linear controllability for nonlinear dynamical networks is a practice that requires extreme care. We have removed the statement “If not careful, such claims

could give the impression that linear network controllability theories are omnipotent [39, 40] . . .” and modified other statements to

- Our detailed comparison between the control importance ranking in a type of biologically meaningful nonlinear control and in linear control for a large number of real pollinator-plant mutualistic networks and a gene regulatory network provides evidence that linear controllability may generate results that are not consistent with nonlinear dynamical behaviors and control of the system. In no way should this be a surprise, as the assumption of linear, time-invariant dynamics cannot be expected to hold for nonlinear dynamical networks in the real world. However, there is a recent tendency to apply the linear controllability framework to real-world nonlinear systems in some specific circumstance such as the *C. elegans* connectome [42] and brain networks [43,44,78,79]. Although the linear control framework may provide insights into nonlinear dynamical networks in some specific circumstances, controlling highly nonlinear dynamical networks is still an open problem at the present. Nonetheless, a thorough analysis of the linear controllability would give clues to its inappropriateness and possible failure in real world systems (see **SI**).

Comment 11 part 2: *“The sole possible exception is for ref [38], which the authors do investigate in their SI. Their main point is that linear controllability fails to differentiate between the node types (3 types of neurons and, separately, muscle cells). This is a valid criticism of the method – it inherently considers only the wiring structure of the network – and it falls to the researcher to identify biologically plausible control strategies. Indeed, the authors of [38] seem to my eye to take this approach. One can argue that a nonlinear approach is more realistic, and potentially more powerful, but in light of the experimentally supported findings reported in [38], it seems disingenuous to say (as the authors do in the SI) that the correlations they have found suggest the linear framework’s ”utter irrelevance to C. elegans connectome.” ”*

Response: We have removed the inappropriate term “utter irrelevance to *C. elegans* connectome” in Supporting Information. However, our analysis generates results that contradict the claim in reference 38 [5] in the original manuscript. We have changed the statement in Supporting Information to

- Specifically, from the point of view of biology, neurons send signals to the muscle cells, but not the other way around. From the standpoint of actual control of the network, a biologically meaningful driver set should favor neurons. Yet the linear controllability theory gives the opposite result, in contrast to the claim in Ref. [1].

Comment 12: *“On page 12, the authors write ”Our finding that almost all nodes can be in some realizations of the minimum controller set with approximately equal probability rules out any possibility of using or exploiting linear controllability as a centrality measure for the purpose of nodal ranking.” While I agree with the broad point here – that there are typically many minimal control sets and many nodes exist in many of them – I think this is perhaps too strong. It is meaningful that some nodes are always or never in a driver set, and the distribution of these nodes in the topology of the network*

is informative and characteristic of empirical context (or the model used to build the network). See the authors' ref 15 and the above-mentioned DOI 10.1038/srep46251, written by some of the same authors.”

Response: We agree with the reviewer and have changed the statements (on page 13) accordingly to

- Is it possible to use linear controllability as a kind of centrality measure for complex networks? The answer is “it depends.” An essential requirement for such a measure is the ability to distinguish and rank the nodes in the network according to some criteria [80]. Intuitively, one would hope that the nodes in the minimum controller set may be special and bear importance relative to other nodes. However, as demonstrated in our work, in a complex mutualistic network, the minimum controller set can be anything but unique. For a network of reasonable size, there is typically a vast number of equivalent configurations or realizations of the set, a fact that was seldom stated or studied in the existing literature of linear controllability of complex networks. We note that, besides the linear structural [6] and exact [11] controllability theories, there are alternative frameworks such as the energy or linear Gramian based controllability [76]. However, the Gramian matrix depends on the chosen minimum controller set and the control signal input matrix. Our finding that, for some networks, almost all nodes can be in some realizations of the minimum controller set with approximately equal probability makes it difficult to use or exploit linear controllability as a centrality measure for nodal ranking, such as network A in Fig. 2(e). However, for other networks, some nodes are always or never in a driver set, which give a distribution of network nodes in the minimum controller set. The distribution of these nodes in the topology of the network is informative and characteristic of empirical context [15, 73].

Comment 13: *“The authors normalize their rankings according to the minimum and maximum thresholds observed for a given network. It seems to me that this ignores the range of values taken in a particular network: in one network the thresholds may be clustered near a low value, while in others they are clustered near a high value. It seems to me that something may be lost here, especially when comparing between the linear and nonlinear frameworks for a single network: a broader range of values on an absolute scale makes the reported correlation more meaningful than if the range is very small.*

While I do not expect this to dramatically impact the main results of the paper, it would be useful for the authors to provide some additional detail here.”

Response: We have done calculations and provided a new figure demonstrating the nodal importance rankings without any normalization (Fig. S5 in Supporting Information). The result is essentially the same as that with normalization.

Minor Comment 1: *“This is a quibble, but the sentence ”Given a real world nonlinear network, we focus on the concrete problem of harnessing tipping point dynamics” strikes me as something of a contradiction in terms: a network is a model of a real-world system, with all of the trade-offs and design decisions that come along with the term ”model.” I suggest revision here.”*

Response: We have changed the statement to

- Our approach and main result can be described, as follows. Given a nonlinear dynamical network with its structure determined from empirical data, we focus on the concrete problem of harnessing a tipping point of the system. Generally, a tipping point is a point of no return at which the system transitions from a normal state to a catastrophic state (e.g., massive extinction) or from a catastrophic state to a normal state, in an abrupt manner, as a system parameter changes through a critical point [57, 61, 65-72].

Minor Comment 2: *“The final sentence before the Results section is wordy and should be rewritten.”*

Response: We have rewritten the sentence as

- However, the main point of the present study is to examine the suitability of linear controllability theory for nonlinear dynamical networks.

Minor Comment 3: *“There is an unpaired left parentheses near the end of page 8: ”(one corresponding...””*

Response: We have corrected it.

Minor Comment 4: *“In the Methods, the statement ”In the presence of control, a full recovery of all species abundances can be achieved (the phenomenon of control enabled recovery)” should be supported with a citation.”*

Response: We observed this phenomenon numerically. We have stated this in the revised manuscript.

References

- [1] Tang, E. & Bassett, D. S. Colloquium: Control of dynamics in brain networks. *Rev. Mod. Phys.* **90**, 031003 (2018).
- [2] Tang, E. *et al.* Developmental increases in white matter network controllability support a growing diversity of brain dynamics. *Nat. Commun.* **8**, 1252 (2017).
- [3] Gu, S. *et al.* Controllability of structural brain networks. *Nat. Commun.* **6**, 8414 (2015).

- [4] Muldoon, S. F. *et al.* Stimulation-based control of dynamic brain networks. *PLOS Comput. Biol.* **12**, e1005076 (2016).
- [5] Yan, G. *et al.* Network control principles predict neuron function in the caenorhabditis elegans connectome. *Nature* **550**, 519 (2017).

REVIEWERS' COMMENTS:

Reviewer #1 (Remarks to the Author):

In the revised version all points of the reviewers have been considered satisfactory. I can now finally recommend this ms for publication in Nat Comms.

Reviewer #3 (Remarks to the Author):

Dear Editor & Authors,

In their revisions to ms NCOMMS-19-13922A, the authors have provided a thorough response to the comments from my initial review. I commend the authors for their thoughtful responses; I now recommend the ms be accepted for publication.

Point-by-point response to reviewer comments

Reviewer #1

General Comment: *“In the revised version all points of the reviewers have been considered satisfactory. I can now finally recommend this ms for publication in Nat Comms.”*

Response: We are grateful that the reviewer recommended our paper for *Nature Communications*.

Referee #3

General Comment: *“In their revisions to ms NCOMMS-19-13922A, the authors have provided a thorough response to the comments from my initial review. I commend the authors for their thoughtful responses; I now recommend the ms be accepted for publication.”*

Response: We thank the reviewer for recommending acceptance of our paper.